# Microtubule Destabilizing Sulfonamides as an Alternative to Taxane-Based Chemotherapy

**DOI:** 10.3390/ijms22041907

**Published:** 2021-02-14

**Authors:** Myriam González, María Ovejero-Sánchez, Alba Vicente-Blázquez, Raquel Álvarez, Ana B. Herrero, Manuel Medarde, Rogelio González-Sarmiento, Rafael Peláez

**Affiliations:** 1Laboratorio de Química Orgánica y Farmacéutica, Departamento de Ciencias Farmacéuticas, Facultad de Farmacia, Universidad de Salamanca, 37007 Salamanca, Spain; mygondi@usal.es (M.G.); avicenteblazquez@usal.es (A.V.-B.); raquelalvarez@usal.es (R.Á.); medarde@usal.es (M.M.); 2Instituto de Investigación Biomédica de Salamanca (IBSAL), Hospital Universitario de Salamanca, 37007 Salamanca, Spain; maria.os@usal.es (M.O.-S.); anah@usal.es (A.B.H.); 3Centro de Investigación de Enfermedades Tropicales de la Universidad de Salamanca (CIETUS), Facultad de Farmacia, Universidad de Salamanca, 37007 Salamanca, Spain; 4Unidad de Medicina Molecular, Departamento de Medicina, Facultad de Medicina, Universidad de Salamanca, 37007 Salamanca, Spain; 5Laboratorio de Diagnóstico en Cáncer Hereditario, Laboratorio 14, Centro de Investigación del Cáncer, Universidad de Salamanca-CSIC, 37007 Salamanca, Spain

**Keywords:** tubulin, sulfonamide, antitumor, taxane, combretastatin A-4, breast cancer, gynecologic cancer

## Abstract

Pan-Gyn cancers entail 1 in 5 cancer cases worldwide, breast cancer being the most commonly diagnosed and responsible for most cancer deaths in women. The high incidence and mortality of these malignancies, together with the handicaps of taxanes—first-line treatments—turn the development of alternative therapeutics into an urgency. Taxanes exhibit low water solubility that require formulations that involve side effects. These drugs are often associated with dose-limiting toxicities and with the appearance of multi-drug resistance (MDR). Here, we propose targeting tubulin with compounds directed to the colchicine site, as their smaller size offer pharmacokinetic advantages and make them less prone to MDR efflux. We have prepared 52 new Microtubule Destabilizing Sulfonamides (MDS) that mostly avoid MDR-mediated resistance and with improved aqueous solubility. The most potent compounds, *N*-methyl-*N*-(3,4,5-trimethoxyphenyl-4-methylaminobenzenesulfonamide **38**, *N*-methyl-*N*-(3,4,5-trimethoxyphenyl-4-methoxy-3-aminobenzenesulfonamide **42**, and *N*-benzyl-*N*-(3,4,5-trimethoxyphenyl-4-methoxy-3-aminobenzenesulfonamide **45** show nanomolar antiproliferative potencies against ovarian, breast, and cervix carcinoma cells, similar or even better than paclitaxel. Compounds behave as tubulin-binding agents, causing an evident disruption of the microtubule network, in vitro Tubulin Polymerization Inhibition (TPI), and mitotic catastrophe followed by apoptosis. Our results suggest that these novel MDS may be promising alternatives to taxane-based chemotherapy in chemoresistant Pan-Gyn cancers.

## 1. Introduction

Gynecologic (i.e., ovarian, uterine, vulva, and vagina) and women breast cancer share relevant similarities at the molecular level that cluster them together (Pan-Gyn) and distinguish them from other tumor types [1,2]. According to GLOBOCAN estimates, Pan-Gyn cancers accounted for 19% of new diagnoses worldwide in 2018 and entailed 40% of new cancer cases among females [3]. 39% of women with Pan-Gyn cancers died from them, representing 30% of all female cancer deaths and 13% of total cancer deaths globally. The high incidence and mortality of female breast [4,5], ovarian [6,7], and uterine [8,9] cancers and the acquired resistance to first-line treatments highlight the need for new therapeutic alternatives.

Since the FDA approved the use of Taxol^®^ for advanced ovarian carcinoma in 1992 and metastatic breast cancer in 1994 [10], taxane-based chemotherapy has become the basis for treating breast and ovarian cancers [11]. First-line chemotherapy involves a platinum-taxane combination for ovarian cancer [12,13,14], preferably carboplatin/paclitaxel [15], and taxanes sequenced or combined with anthracyclines for advanced and metastatic breast cancer [16,17,18,19]. Furthermore, the combination of cisplatin and paclitaxel is considered a standard regimen in the palliative setting for patients with metastatic, recurrent, or persistent uterine cancer [9,20,21,22]. 

However, the clinical efficacy of taxanes is limited by an intrinsic low aqueous solubility (<2 µg/mL [23,24,25]), dose-limiting toxicities, and the development of drug resistance [26]. The low solubility (remaining challenging) requires complex solvent-system formulations that can alter the pharmacokinetic profiles [27,28] and have been associated with toxicity [29,30]. The development of resistance to these drugs is usually ascribed to Pgp-mediated efflux and alterations in tubulin (mutations in the tubulin genes, altered tubulin isotype expression, or aberrant levels of tubulin) [31,32,33]. Specific resistance to taxane-based chemotherapy has been described for female breast cancer, with an approximately 30% of recurrence [34,35], ovarian cancer, in which 50% of patients relapse with the chemoresistant disease within the first two years post-diagnosis [36,37], and uterine cancer [38].

Taxanes bind to microtubules, stabilizing lateral contacts between protofilaments, thus promoting assembly and inhibiting depolymerization at high concentrations. At lower concentrations, they alter microtubule dynamics and preclude cells from properly completing mitosis [39,40], which leads to cell cycle arrest at G_2_/M and, subsequently, to cell death mainly triggered by apoptosis [41,42]. There are other drug binding sites in tubulin that affect microtubule dynamics similarly to taxanes, but whose engagement results in microtubule depolymerization at high concentrations, such as the colchicine, the vinca alkaloids, the maytansine, the pironetin, and the hemiasterlin sites [43]. The only approved drugs belong to the vinca alkaloids site binding ligands, which are used against hematological malignancies and solid tumors, including breast cancer. However, they share with taxanes the problems of toxic side effects, difficult pharmacokinetics, and complex hydrophobic structures that make them substrates of drug efflux pumps [44]. 

The colchicine site offers advantages for drug design since its ligands exhibit simple chemical structures and are synthetically accessible. Colchicine-site ligands display antiproliferative effects (characterized by cell cycle arrest at G_2_/M phase followed by the induction of apoptosis [45]) and act as vascular disrupting agents, affording additional odds against solid tumors [46]. Representatives such as the stilbene combretastatin A-4 (CA-4) and the sulfonamide ABT-751 have reached clinical trials [47]. However, the *Z* olefin of combretastatins is chemically unstable and readily isomerizes to the more stable but inactive *E* isomer. The low solubility of CA-4 requires the use of phosphate prodrugs on the phenolic hydroxyl group, which is, in turn, the target for metabolic inactivation by glucuronidation in resistant cells, such as the colon adenocarcinoma HT-29 cell line [48]. ABT-751 is an orally administered sulfonamide with modest potency against human cancer cell lines and xenograft models. Despite its favorable pharmacokinetics, ABT-751 has not found clinical application due to insufficient potency [49]. 

In this work, we have designed and synthesized a new family of Microtubule Destabilizing Sulfonamides (MDS) hybrids of CA-4 and ABT-751. The effects of replacing the chemically unstable CA-4 olefin with a sulfonamide bridge, the removal or substitution of the phenolic hydroxyl group, and the introduction of several modifications on the aromatic rings and the sulfonamide bridge have been explored while maintaining the 3,4,5-trimethoxyphenyl ring that has been long considered essential for high potency [50,51] (Figure 1). The resulting compounds have been evaluated for tubulin inhibition in vitro and antiproliferative activity against several human tumor cell lines. We have also studied whether MDR1 pumps could compromise their effectiveness by the pharmacological inhibition of Pgp using verapamil. After a comprehensive preliminary evaluation, three promising MDS have been further screened against several representative cancer cell lines representative of the tumor types that are associated with the highest mortalities: breast, ovarian, and uterine, accounting for 51%, 15%, and 32% of cancer deaths in women, respectively. The effect of the compounds on cancer cell proliferation has been studied and compared with paclitaxel, CA-4, and ABT-751. The mechanism of action of these novel MDS has been studied by ascertaining their effect on the microtubule network in vitro. These MDS induce mitotic arrest, followed by apoptotic cell death with differences arising from different genetic backgrounds of the studied cell lines. The favorable pharmacodynamic and pharmacokinetic profiles compared to reference drugs, including solubility, absence of Pgp-mediated resistance, and improved potency indicate that MDS are promising candidates for the treatment of this kind of malignancies.

## 2. Results

### 2.1. Synthesis of MDS

52 new MDS (Figure 1) were prepared following the synthetic approach shown in Figure 2 (detailed synthetic procedures and NMR spectra can be found in Appendix A). The synthesized compounds were divided into three series according to the substituents on the aromatic B ring (Ar_B_): series 1 (compounds **1a**-**24**), series 2 (**25**–**38**), and series 3 (**39**–**48b**) (Table 1). Sulfonamides were built up by the reaction between 4-methoxy- (series 1), 4-nitro- (series 2), or 4-methoxy-3-nitro- (series 3) benzenesulfonyl chlorides and 3,4,5-trimethoxyaniline, providing crystalline products in excellent yields (90–96%). Nitro groups were reduced to amines by palladium-catalyzed hydrogenation (82–98% yields). The subsequent amino derivatization by alkylation, acylation, and/or formylation-reduction sequences allowed the introduction of varied *Z*, *R*, and *Y* substituents (Figure 1). Substitutions at the sulfonamide nitrogen were conducted by alkylation reactions with alkyl halides in KOH/CH_3_CN (methylations with methyl iodide in 63–98% yields) or K_2_CO_3_/DMF (ethyl, acetyl, acetonitrile, benzyl, or ethyl acetate substituents in 40–99% yields). 

### 2.2. Replacement of CA-4 Olefin by a Sulfonamide Highly Increased Aqueous Solubility

Aqueous solubility of representative compounds was spectrophotometrically measured in pH 7.0 phosphate buffer (Table 2). UV absorbance at three maximum absorbance wavelengths of selected compounds was measured after 48 h shaking and subsequent microfiltration. Replacement of the CA-4 olefinic bridge by a sulfonamide group highly increased aqueous solubility when compared to paclitaxel and to CA-4 (4- up to 1700-fold increase). Similar or even better results were obtained when compared with the orally administered sulfonamide ABT-751 (up to 42-fold increase). The best results were achieved by the introduction of amino acid derivatives at the *Z* position (compounds **21** and **22**).

### 2.3. MDS Inhibit Cell Proliferation in Breast, Cervix, and Ovarian Tumor Cells

The in vitro antiproliferative activity of MDS at 1 µM after 72 h treatments was evaluated by a colorimetric method against the human tumor cell lines HeLa (cervix epithelioid carcinoma) and MCF7 (breast adenocarcinoma). Compounds showing more than 40% inhibition at 1 µM in at least three independent assays were evaluated in a range of concentrations between 0.1 nM and 10 µM for the determination of half-maximal inhibitory concentrations (IC_50_) (Table 1). Paclitaxel [52], CA-4, and ABT-751 were used as references.

HeLa and MCF7 cells showed similar sensitivities to MDS, both cell lines displayed less than two-fold changes in IC_50_ values between them in most cases. The most potent compounds (e.g., **29a**, **30**, **38**, **42**, **43**, **44**, and **45**) showed IC_50_ values in the double-digit nanomolar range, less potent than paclitaxel and CA-4 but more than ABT-751. Substitutions on the trimethoxyphenyl ring resulted in inactive compounds (**11–24**). Alkylations of the sulfonamide improved the potency, with methyl groups yielding the lowest IC_50_ values in most cases (e.g., **29a**, **30**, and **42**). Larger substituents gave worse results except for some benzyl (e.g., **45**) and cyanomethyl (e.g., **44**) derivatives. Replacement of the 4-methoxy group on the B ring in series 1 by an amino substituent (series 2) and introduction of an amino group on position 3 as a primary amine (series 3) gave favorable results. Substituents with a hydrogen bond donor, such as methylamines (**38**) in series 2, or primary amines in series 3 (**42–45**), resulted in the most potent MDS. Based on antiproliferative results, we selected 3 out of the 7 most potent compounds (**38**, **42**, and **45**) as representative leaders for further investigation. 

The in vitro antiproliferative activity of compounds **38**, **42**, and **45** against a panel of five human ovarian carcinoma cell lines (SKOV3, IGROV-1, A2780, OVCAR-8, and OVCAR-3) was evaluated by a colorimetric method (Table 3) and compared with paclitaxel [14]. Paclitaxel and MDS showed a similar comparative behavior against the highly paclitaxel-sensitive A2780 and OVCAR-8 and the less sensitive IGROV-1 cell line, with paclitaxel being one order of magnitude more potent than MDS. Concerning OVCAR-3, paclitaxel is less potent, but MDS maintain their potency range, thus resulting in similar effects of both types of drugs. For SKOV3, the potency reduction observed for paclitaxel together with a sustained or improved (**42** with a 7 nM IC_50_ value) potency for MDS results in a more favorable profile of the MDS compared to paclitaxel.

### 2.4. Lead Compounds Overcome MDR-Mediated Resistance

The antiproliferative activity of MDS against the CA-4-resistant cell line HT-29 (colon adenocarcinoma) in the absence or the presence of 10 µM verapamil, a non-selective Pgp/MDR1 inhibitor, was measured 72 h post-treatment by a colorimetric method (Appendix A). Verapamil itself did not affect cell proliferation at that concentration. Detecting changes in drug potency when inhibiting MDR-mediated efflux reflect that these transporters reduce the intracellular drug concentration. HT-29 has been reported to express moderate levels of MDR1, accounting for a resistant phenotype [53]. A comparison of the antiproliferative IC_50_ values of each MDS in the presence or absence of verapamil provides an estimate of the susceptibility of each compound to efflux pumps. Most of the compounds, including leading derivatives **38**, **42**, and **45** (Table 4 and Appendix A), showed no significant differences (up to two-fold change) between pairs of IC_50_ values, thus suggesting that they are not substrates of MDR pumps. Compounds **3**, **29a**, and **30** showed higher potencies against HT-29 cells when combined with verapamil (comparables to that of obtained against sensitive HeLa an MCF7 cells), thus validating the assay as positive controls.

### 2.5. Lead Compounds Induce G_2_/M Arrest in Breast, Ovarian and Cervix Tumor Cells 

The effect of the lead MDS on the cell cycle of MCF7, SKOV3, and HeLa cells was analyzed in time-course experiments at 24, 48, and 72 h post-treatment by DNA staining and flow cytometry quantification. The working concentration for each compound (600 nM for **38**, 50 nM for **42**, and 400 nM for **45**) was established at the complete inhibition of cell proliferation in view of individually assays run in parallel for each cell line.

For breast adenocarcinoma MCF7 cells (Figure 3a) a pronounced and sustained G_2_/M arrest was observed for all the treatments from 24 h (70–80%) to 72 h (60–76%), compared to untreated MCF7 cells, which maintained a 32–40% population at G_2_/M throughout the time course. A moderate increase of the percentage of cells at the SubG_0_/G_1_ region was observed, which slightly increased over time (from 4–6% at 24 h to 6–12% at 72 h) accompanied by the decrease in the G_2_/M population.

Compounds **38**, **42**, and **45** (Figure 3b) arrested ovarian carcinoma SKOV3 cells at the G_2_/M phase 24 h post-treatment (44%) compared to untreated cells (30%). The percentage of SubG_0_/G_1_ treated SKOV3 cells was already elevated at 24 h (20% to 22%) compared to untreated controls (always <14%). Treatments with compounds **38** and **42** increased the percentage of SubG_0_/G_1_ modestly at 48 h (35% and 34%, respectively), and notably at 72 h (43% and 53%, respectively). This increase presumably occurred at the expense of cells arrested at the G_2_/M phase. The effect of compound **45** was notably higher, with 46% of cells at SubG_0_/G_1_ region 48 h post-treatment and 63% at 72 h. 

Treatment of HeLa cells with **38** and **45** (Figure 3c) caused a significant arrest at the G_2_/M phase after 24 h (80%) and a more modest one for compound **42** (44%) accompanied by a limited increase in the percentage of cells of the SubG_0_/G_1_ population (9–16% vs. <4% in untreated controls). At subsequent time points, around 50% of the cells for all three treatments were allocated to the SubG_0_/G_1_ region of the cell cycle histogram at the expense of the G_2_/M phase.

### 2.6. Lead Compounds Trigger Apoptotic Cell Death

To determine whether the cytotoxic/antiproliferative effect of the MDS was due to apoptosis induction, cells were treated with variable concentrations of the drug for 72 h or 6 days, stained with Annexin V-FITC (AnV) and propidium iodide (PI), and analyzed by flow cytometry. Double negative cells (AnV^−^/PI^−^) were classified as live cells (L), cells positive for AnV and negative for PI (AnV^+^/PI^−^) were considered in Early Apoptosis (EA), cells negative for AnV and positive for PI (AnV^−^/PI^+^) were considered necrotic (N), and double-positive cells (AnV^+^/PI^+^) as Late Apoptosis (LA). In all cases, the compounds were used at the same concentrations previously used in cell cycle studies (Table 5).

Breast cancer MCF7 cells treated with compound **42** for 72 h were mostly classified as live cells (88%), whereas lower percentages of live cells were found when treated with compounds **38** (60%), **45** (71%) and the reference compound CA-4 (58%). This lower percentage of live cells was accompanied by a higher proportion of EA cells for **38** (29%), **45** (20%), and CA-4 (32%) compared to the treatment with **42** (9%). Longer treatment exposures of 6 days confirmed this trend, with **38**, **45**, and CA-4 showing significantly lower percentages of live cells (45–57%) and higher EA populations (39–52%) than **42**, which showed live-cell percentages of 79% and EA of 19%.

Similar results were found 72 h post-treatment for SKOV3 ovarian tumor cells for MCF7 cells. SKOV3 cells treated with compounds **38** and **45** showed modest percentages of EA cells (31% and 23%, respectively) whereas a lower apoptotic population was found for compound **55** (12%). These percentages of apoptotic cells for compounds **38** and **55** were accompanied by low percentages of live cells (62% and 73%, respectively), with 10 nM paclitaxel showing high percentages of live cells (91%).

Treatment of the cervical cancer HeLa cells with compounds **38**, **42**, and **45** resulted in a severe reduction of live cells down to less than 7% and an accumulation of LA (82–88%) and necrotic (9–13%) cells after 72 h.

### 2.7. Lead Compounds Disrupt the Microtubule Network and Inhibit Tubulin Polymerization

The effect of MDS **38** and **45** on the tubulin cytoskeleton was analyzed in MCF7, SKOV3, and HeLa cells after 72-h treatments. The microtubule network was visualized by confocal microscopy staining α-tubulin in green, and cell nuclei were stained in blue with DAPI (4’,6-diamidino-2-phenylindole) (Figure 4). Compounds **38** and **45** disrupted the microtubule system of MCF7, SKOV3, and HeLa cells (Figure 4a), which loses the hairy appearance of untreated cells and shows up as a diffuse gauze. Treated cells showed multilobulated nuclei resembling bunches of grapes (Figure 4b).

The effect of all the synthesized compounds on the in vitro tubulin polymerization was evaluated at 10 µM. All the compounds that inhibited polymerization more than 30% were also assayed at 20 µM. IC_50_ values were calculated for those compounds inhibiting tubulin polymerization by at least 50% at 20 µM (Table 1 and Appendix A). Most antiproliferative *N*-alkylated sulfonamides interfere to a greater or lesser extent with tubulin polymerization, with the highest effect shown by compound **45** (IC_50_ = 3.7 µM), with a benzyl group on the sulfonamide nitrogen, close to CA-4 (IC_50_ = 3 µM) and better than ABT-751 (IC_50_ = 4.4 µM). Compounds **38** and **42** showed moderate to good IC_50_ values of 10 and 12 µM, respectively. 

The binding mode of the active compounds at the colchicine site of tubulin was studied by flexible docking studies. The protein conformational variability was represented by using 55 protein structures with different binding sites, as previously described [54,55]. Two frequently used docking programs with different scoring functions were used for the flexible docking of the ligands in the binding sites. Several thousand poses were generated for each ligand which were automatically classified according to their occupation of the subsites of the colchicine domain, and the poses with the best consensus scores were selected (Figure 5). The selected ligands bind in similar dispositions to combretastatin A-4, with a good overlap of the trimethoxyphenyl ring, which is inserted edgewise towards the surface of sheets S8 and S9 between the sidechains of Ala316β, Val318β, Ala354β, and covered by helices H7 and H8 and the H7-H8 loop and interacting with Cys241β, Leu242β, Leu248β, Ala250β, and Leu255β. The sulfonamide bridges are placed close to the olefin binding zone of CA-4, with the other aromatic ring behind helix H8 and above the sidechains of Ala316β and the methylenes of Lys352β, with the substituents on the sulfonamide nitrogen projecting towards a hydrophobic region of the interdimer interface, thus explaining the preference for small hydrophobic substituents such as methyl or cyanomethyl groups.

## 3. Discussion

Taxanes—usually paclitaxel alone or in combination regimens—entail the first-line chemotherapy for breast and gynecological cancers [9,13,18,20]. However, limitations related to toxicity, pharmacokinetic problems, and the appearance of resistance [31] prompt the search for new treatments. This work has yielded a new family of tubulin inhibitors with a sulfonamide scaffold as hybrids of combretastatin A-4 and ABT-751, two colchicine site ligands in clinical trials that present shortcomings of opposite nature whose hybridization might alleviate (Figure 1). 

Therefore, a focused library of sulfonamides was designed, and 52 representatives were synthesized following a synthetic methodology that involves first the assembly of the sulfonamide followed by functional group manipulations at different points of the main scaffold. A breadth of substituents was thus introduced to explore the chemical space for optimal pharmacokinetic and pharmacodynamic properties (Figure 1). One of the main drawbacks of taxanes and CA-4 is their low water solubility, making necessary drug administration as formulations with several associated adverse effects [27,28,47]. New sulfonamide derivatives show moderate to good solubilities (Table 2). Improvements up to 1700 times in solubility were found when compared with reference drugs paclitaxel and CA-4 and even surpass the solubility of the orally administered ABT-751 up to 42 times.

The antiproliferative activity of the synthesized compounds was evaluated against two Pan-Gyn representative human tumor cell lines (HeLa and MCF7) (Table 1). Many compounds showed growth inhibitory activities at submicromolar concentrations, thus validating the adequacy of the selected diarylsulfonamide scaffold. Most potent derivatives inhibited cell proliferation at very low drug concentrations, with IC_50_ values in the low to medium nanomolar range. Compounds **2a**, **29a**, **30**, **32**, **38**, and **42–45** display antiproliferative IC_50_ values comparable to those of paclitaxel and CA-4 and better than ABT-751, a tubulin colchicine-site inhibiting sulfonamide in clinical trials used as a reference [49]. Alkylation of the sulfonamide nitrogen significantly increases potency compared to the unsubstituted sulfonamides, probably due to a combination of a favoring of the *cisoid* disposition of both aromatic rings, an essential requirement for colchicine-site-binding drugs, with a more favorable interaction with the target [56]. Small alkyl groups, such as methyl or ethyl (e.g., **2a**, **29a**, **38**, or **42**), are usually preferred over longer chains such as carboxylic acid derivatives. All the modifications attempted on the trimethoxyphenyl ring abolished the activity, whereas the introduction of hydrogen bond donor amines on the B ring translated into more potent analogs, such as the secondary amines in the *para* position of the sulfonamide group (**38**) and the primary amines *ortho* to the methoxy group of B ring (**42–45**). Both modifications improve the polarity and aqueous solubility. The potency improvement suggests favorable hydrogen bonding with the target. Thus, B-ring amines improve both the pharmacokinetics and the pharmacodynamics.

The antiproliferative IC_50_ values of each compound against HeLa and MFC7 cell lines were quite similar, whereas the CA-4-resistant HT-29 cell line was less or at best equally sensitive to MDS (Appendix A). When differences in antiproliferative potencies occur, they are not due to multidrug-resistance by the described moderate expression of MDR1 or MDR3 proteins in HT-29 colon carcinoma cell line [48,53,57], as only a handful of compounds experienced a potency increase in the presence of the Ppg/MDR1 inhibitor, verapamil [58]. Therefore, none of our lead compounds (**38**, **42**, and **45**) seemed to be MDR substrates and even compounds **38** and **42** overcame the resistance shown by CA-4 in the HT-29 cell line (IC_50_ = 59, 81, and 305 nM, respectively) (Table 4).

The microtubule disruption caused by the lead compounds in MCF7 cells winded up in the accumulation of cells in the G_2_/M region (Figure 3a), as previously reported for CA-4 and ABT-751, used as references [59,60]. This sustained mitotic arrest lasted at least 72 h with percentages of cells showing DNA fragmentation below 12%. The mild response regarding this apoptotic marker is in line with the results observed in AnV/PI staining experiments (Table 5). Compounds **38** and **45** triggered phosphatidylserine exposure in 28–34% of MCF7 cells; 11% in the case of **42**. The lack of PI staining within the apoptotic population reveals the integrity of the plasma membrane in most Annexin V-positive cells. This slight apoptotic response might be related to the expression levels of caspase 3, the hub effector of the apoptotic machinery. MCF7 cells have been reported to be deficient in caspase 3 [61,62]. Despite MCF7 cells suffer a weak apoptotic response upon treatment with the lead compounds for 72 h, the percentage of early apoptotic cells increases after 6-day treatments, still preserving the integrity of the plasma membrane. This slow build-up of the apoptotic response in MCF7 cells may putatively rely on the other two effector caspases 6 and 7, as described for DNA damaging agents [11,63,64].

Lead compounds **38**, **42**, and **45** were then assayed against a panel of human ovarian tumor cell lines. Ovarian serous adenocarcinoma cell lines (SKOV3, OVCAR-8, and OVCAR-3), the most common type of ovarian cancer, were more sensitive than endometrioid cell lines (IGROV-1 and A2780). The metastatic cell line SKOV3 was found quite resistant to paclitaxel and CA-4 (IC_50_ of 81 and 380 nM [65], respectively) but was sensitive to MDS (Table 3). Time-course analysis of the cell cycle distribution (Figure 3b) showed a strong mitotic arrest for **42**, **38, and 45** 24 h after treatment. The mitotic arrest readily triggered apoptosis 48 h after treatment. Compound **45** elicits a higher SubG_0_/G_1_ population than **38** and **42** at 72 h. Dual staining experiments confirm the apoptotic response with Annexin V-positive cells at 72 h ranging from 14–36%, notably higher when compared with paclitaxel (7%).

Studies on the mechanism of action of MDS on the cervical cancer cell line HeLa were also performed (Figure 3c). After the common mitotic arrest at G_2_/M phase 24 h after treatment (more pronounced for **38** and **45** than for **42**) HeLa cells exhibited strikingly high percentages of SubG_0_/G_1_ cells (around 50%) from 48 h after drug exposure. AnV/PI staining experiments revealed a strong and advanced apoptotic response with rates of double-positives cells above 82%, indicating the permeabilization of the plasma membrane.

At this point, we wished to verify if the observed effect was due to the disruption of microtubule assembly. Tubulin immunofluorescence analysis of cells treated with compounds **38** and **45** revealed a clear disruption in the microtubule network (Figure 4a). Furthermore, as interference with microtubule dynamics is expected to affect chromatin organization, cell nuclei were stained with DAPI. Accumulation of multiple micronuclei characteristic of mitotic catastrophe was observed, widely linked to tubulin-binding drugs [66,67]. In vitro tubulin polymerization inhibition experiments confirmed the effect of the compounds on tubulin, and molecular docking experiments suggest a similar binding mode to that of CA-4 to the colchicine site of tubulin (Figure 5). However, no correlation was observed between TPI and antiproliferative IC_50_ values, as has previously been found for many tubulin polymerization inhibitors [68]. This discrepancy can be explained by an antiproliferative effect dependent on the alteration of microtubule dynamics and not on polymer mass, as measured in TPI assays.

In conclusion, a new family of MDS has been designed and synthesized. MDS antiproliferative effects against several representative cell lines of Pan-Gyn cancers have been studied. The effects of the compounds vary depending on the structural modifications, the most potent derivatives being those with a small alkyl substituent on the sulfonamide nitrogen. The molecular mechanism of action studies revealed a pronounced G_2_/M arrest mainly followed by a SubG_0_/G_1_ increase over time. The different behavior between MCF7, SKOV3 and HeLa cells seems to be related to the rate at which each cell line builds up its apoptotic response to the initial unsolved mitotic arrest. Inhibition of tubulin polymerization in vitro was observed for most potent MDS, as well as disruption of microtubule network by immunofluorescence, probably by binding similarly to CA-4 to the colchicine site of tubulin. The improved aqueous solubility, lack of Pgp-mediated resistance, and antiproliferative potencies comparable to paclitaxel—first-line treatment of Pan-Gyn cancers—make MDS interesting for future application as antitumor agents.

## 4. Materials and Methods

### 4.1. Chemistry

#### 4.1.1. Chemical Synthesis

Detailed synthetic procedures and characterization of compounds can be found in Appendix A

#### 4.1.2. Chemical Characterization of Lead MDS **38**, **42**, and **45**

**MDS 38.** M.p.: 147–150 °C (CH_2_Cl_2_/Hexane). IR (KBr): 3390, 1599, 1502, 835 cm^−1^. ^1^H NMR (400 MHz, CDCl_3_): δ 2.83 (3H, *s*), 3.07 (3H, *s*), 3.71 (6H, *s*), 3.80 (3H, *s*), 6.28 (2H, *s*), 6.51 (2H, *d, J* = 8.8), 7.35 (2H, *d, J* = 8.8). ^13^C NMR (100 MHz, CDCl_3_): δ 30.0 (CH_3_), 38.4 (CH_3_), 56.1 (2CH_3_), 60.8 (CH_3_), 104.6 (2CH), 110.8 (2CH), 122.6 (C), 130.0 (2CH), 137.2 (C), 137.8 (C), 152.7 (C), 152.8 (2C). HRMS (C_17_H_22_N_2_O_5_S + H^+^): calcd 367.1322 (M + H^+^), found 367.1325.

**MDS 42.** M.p.: 124–126 °C (MeOH). ^1^H NMR (400 MHz, CDCl_3_): δ 3.11 (3H, *s*), 3.73 (6H, *s*), 3.83 (3H, *s*), 3.90 (3H, *s*), 3.94 (2H, *s*), 6.30 (2H, *s*), 6.78 (1H, *d, J* = 8.8), 6.90 (1H, *d, J* = 2.4), 6.99 (1H, *dd, J* = 8.8 and 2.4). ^13^C NMR (50 MHz, CDCl_3_): δ 38.5 (CH_3_), 55.7 (CH_3_), 56.0 (2CH_3_), 60.8 (CH_3_), 104.6 (2CH), 109.1 (CH), 113.2 (CH), 118.8 (CH), 128.0 (C), 136.5 (C), 137.3 (C), 137.7 (C), 150.2 (C), 152.8 (2C). HRMS (C_17_H_22_N_2_O_6_S + H^+^): calcd 383.1271 (M + H^+^), found 383.1263.

**MDS 45.**^1^H NMR (200 MHz, CDCl_3_): δ 3.63 (6H, *s*), 3.78 (3H, *s*), 3.93 (3H, *s*), 4.63 (2H, *s*), 6.14 (2H, *s*), 6.83 (1H, *d, J* = 8.4), 7.02 (1H, *d, J* = 2), 7.11 (1H, *dd, J* = 8.4 and 2), 7.23 (5H, *bs*). ^13^C NMR (100 MHz, CDCl_3_): δ 55.3 (CH_2_), 55.8 (CH_3_), 55.9 (2CH_3_), 60.8 (CH_3_), 106.7 (2CH), 109.3 (CH), 113.1 (CH), 118.8 (CH), 127.6 (CH), 128.3 (2CH), 128.7 (2CH), 130.5 (C), 134.9 (C), 136.2 (C), 136.6 (C), 137.6 (C), 150.3 (C), 152.7 (2C). HRMS (C_23_H_26_N_2_O_6_S + H^+^): calcd 459.1584 (M + H^+^), found 459.1589.

#### 4.1.3. Aqueous Solubility

The aqueous solubility of the sulfonamides was determined using an approach based on the saturation shake-flask method. Tested compounds (1–2 mg) were stirred in pH 7.0 phosphate buffer (300 µL) for 48 h at room temperature. The resulting suspension was filtered over a 45 µm filter to discard insoluble residues, and the concentration in the supernatant was measured by UV absorbance. To determine the concentration, a scan between 270 and 400 nm was performed in a Helios-α UV-320 Spectrophotometer (Thermo-Spectronic, Thermo Fischer Scientific, Waltham, MA, USA) for each tested compound. Then, three maximum wavelengths of absorbance per compound were selected from the previous scan and calibration curves were performed at these three wavelengths. Solubility results are given as the average of the three measurements.

### 4.2. Biological Evaluation

#### 4.2.1. Cell Lines and Cell Culture Conditions

MCF7 (human breast carcinoma), SKOV-3 (human ovarian carcinoma), OVCAR-3 (human ovarian carcinoma), OVCAR-8 (human ovarian carcinoma), and HeLa (human cervical carcinoma) cell lines were cultured in Dulbecco’s Modified Eagle’s Medium (DMEM) (Gibco, Thermo Fischer Scientific) containing 10% (*v*/*v*) heat-inactivated fetal bovine serum (HIFBS) (Lonza-Cambrex, Karlskoga, Sweden), 2 mM L-glutamine (Lonza-Cambrex, Karlskoga, Sweden), 100 μg/mL streptomycin, and 100 IU/mL penicillin (Lonza-Cambrex) at 37 °C in humidified 95% air and 5% CO_2_. HT-29 (human colon carcinoma), IGROV-1 (human ovarian carcinoma), and A2780 (human ovarian carcinoma) cell lines were cultured in RPMI 1640 medium (Gibco, Thermo Fischer Scientific) supplemented with 10% HIFBS, 100 μg/mL streptomycin, and 100 IU/mL penicillin at 37 °C in humidified 95% air and 5% CO_2_ atmosphere. The presence of mycoplasma was routinely checked with MycoAlert kit (Lonza-Cambrex) and only mycoplasma-free cells were used in the experiments. Ovarian tumor cell lines were originally acquired from Dr Atanasio Pandiella from Centro de Investigación del Cáncer (CIC), Salamanca, Spain. HeLa, HT-29 and MCF7 tumor cell lines were provided by Dr Faustino Mollinedo from Centro de Investigaciones Biológicas Margarita Salas, Madrid, Spain. 

#### 4.2.2. Cell Proliferation Assay

MCF7, HeLa, and HT-29 cell proliferation, when treated with the corresponding compounds, were determined using the XTT (sodium 3’-[1(phenylaminocarbonyl)-3,4-tetrazolium]-bis(4-methoxy-6-nitro)-benzenesulfonic acid hydrate) cell proliferation kit (Roche Molecular Biochemicals, Mannheim, Germany). A freshly prepared mixture of XTT labeling reagent with 0.02% (*v*/*v*) PMS (*N*-methyl-dibenzopyrazine methyl sulfate) electron coupling reagent was added to cells (50 µL/well in 96-well plates, total volume of 160 µL/well). Cells were incubated under standard culture conditions for 4 h (MCF7 and HeLa) or 6 h (HT-29). The absorbance of the formazan product generated was measured at 450 nm using a multi-well plate reader. SKOV-3, OVCAR-3, OVCAR-8, IGROV-1, A-2780 cell proliferation, when treated with the corresponding compound, were determined using MTT (3-(4,5-dimethylthiazol-2-yl)-2,5-diphenyl-2*H*-tetrazolium bromide) (Sigma-Aldrich, St. Louis, MO, USA). MTT in Phosphate Buffer Saline (PBS) (5 mg/ml) was added to cells (110 µL/well in 24-well plates, total volume of 1110 µL/well). After 1 hour of incubation, the medium was aspirated, and formazan violet crystals were dissolved in dimethyl sulfoxide (DMSO) (500 µL/well). Absorbance was measured at 570 nm in a plate reader (Ultra Evolution, Tecan, Männedorf, Suiza). To determine cell viability, cells in exponential growth phase were seeded (100 µL/well in 96-well plates and 1000 µL/well in 24-well plates) with appropriate cell line concentration (1.5·10^4^ cells/mL for MCF7 and HeLa, 3·10^4^ cells/mL for HT-29, and 1·10^4^ cells/mL for SKOV-3, OVCAR-3, OVCAR-8, IGROV-1, and A-2780) in complete RPMI 1640 or DMEM medium at 37°C and 5% CO_2_ atmosphere. After 24 h incubation, to allow cells to attach to the plates, all compounds were added at 1 µM concentration and the effect on the proliferation was evaluated 72 h post-treatment. Compounds showing antiproliferative effects at tested concentration were selected for IC_50_ calculation (50% inhibitory concentration with respect to the untreated controls) from 10^−5^ to 10^−10^ M concentration. Non-linear curves fitting the experimental data were carried out for each compound. Compounds were dissolved in DMSO and the final solvent concentrations never exceeded 0.5% (*v*/*v*). The control wells included treated cells with 0.5% (*v*/*v*) DMSO and the positive control. 10 µM verapamil was included as a control for the HT-29 cell line. Measurements were performed in triplicate, and each experiment was repeated three times. 

#### 4.2.3. Cell Cycle Analysis

Cell cycle analysis was performed in MCF7, SKOV-3, and HeLa cells by quantifying the DNA content by flow cytometry. Cells in the exponential growth phase at 2·10^4^ cells/mL were seeded in 24-well plates (1 mL/well). After 24 h incubation, cells were treated with different concentrations of the selected compounds and the effect of treated and untreated cells was measured 24, 48, and 72 h post-treatment. Live and dead cells were collected and fixed in ice-cold ethanol/PBS (7:3) and stored at 4 °C for later use. Cells were rehydrated with PBS, treated with 100 µg/mL RNase A (Sigma-Aldrich Co., St. Louis, MO, USA) and stained overnight in darkness at room temperature with 50 µg/mL PI (Sigma-Aldrich Co.). Cell cycle profiles were then analyzed by flow cytometry using BD Accuri™ C6 Plus Flow Cytometer (BD Biosciences, San José, CA, USA). Data were analyzed with BD Accuri™ C6 Software (version 1.0.264.21, BD Biosciences, San José, CA, USA) and compared to control cells. Compounds were dissolved in DMSO and the final solvent concentration never exceeded 0.5% (*v*/*v*). Control wells included cells with 0.5% (*v*/*v*) DMSO.

#### 4.2.4. Apoptotic Cell Death Quantification

MCF7, SKOV-3, and HeLa apoptotic cells were quantified using an Annexin V-FITC/PI apoptosis detection kit (Immunostep, Salamanca, Spain) according to the manufacturer’s guidelines. 1.5 mL/well of cells in the exponential growth phase at 2·10^4^ cells/mL were seeded onto 12-well plates and left to attach overnight. After 24 h incubation, cells were treated with selected compounds. After 72 h (MCF7, SKOV-3, and HeLa) and 6 days (MCF7) post-treatment, attached and floating cells of treated and untreated wells were collected, centrifuged, resuspended in Annexin V binding buffer, and stained with Annexin V-FITC/PI. Cells were then incubated in darkness at room temperature for 15 min and a total of 30,000 cells were acquired and analyzed using the BD Accuri™ C6 Plus Flow Cytometer and Software (version 1.0.264.21, BD Biosciences, San José, CA, USA), respectively. Compounds were dissolved in DMSO and the final solvent concentration never exceeded 0.5% (*v*/*v*). Control wells included cells with 0.5% (*v*/*v*) DMSO.

#### 4.2.5. Immunofluorescence

MCF7, SKOV-3, and HeLa cells were grown on glass coverslips coated with poly-L-lysine (12 mm diameter). To reach appropriate cell confluence, coverslips were manipulated in 6-well plates (3 coverslips/well) seeding 3 mL/well of cells in the exponential growth phase at 2 × 10^4^ cells/mL. After 24 h incubation, cells were treated with selected compounds and incubated for 72 h. Cells were washed with PBS, fixed with 4% formaldehyde in PBS for 10 min, permeabilized with 0.5% Triton X-100 (Boehringer Mannheim, Ingelheim am Rhein, Germany) in PBS for 10 min and blocked with 10% BSA in PBS for 30 min. Microtubule detection was performed by incubation with anti-α-tubulin mouse monoclonal antibody (1:200 in 3% BSA/PBS) (Sigma-Aldrich) for 1.5 h. After PBS washing, coverslips were incubated with fluorescent secondary antibody Alexa Fluor 488 goat anti-mouse IgG (1:400 in 1% BSA/PBS) (Molecular Probes, Invitrogen, Thermo Fischer Scientific) for 1 h in darkness. After being washed with PBS, cell nuclei were stained with DAPI (dihydrochloride of 4′,6-diamidino-2-phenylindole) (diluted 1:10,000 in mq H_2_O) (Roche, Basel, Switzerland) for 5 min in darkness. DAPI excess was removed by washing with PBS. Mowiol reagent (Calbiochem, Sigma-Aldrich) was used to mount preparations on slides. Cells were analyzed by confocal microscopy using a LEICA SP5 microscope DMI-6000V model coupled to a LEICA LAS AF software computer.

#### 4.2.6. Tubulin Isolation

Microtubular protein was isolated from calf brain according to the modified Shelanski method [69,70] by two cycles of temperature-dependent assembly/disassembly and stored at −80 °C. Before each use, protein concentration was determined by the Bradford method [71] taking BSA as standard.

#### 4.2.7. Tubulin Polymerization Inhibition (TPI) Assay

Tubulin polymerization was monitored using a Helios α spectrophotometer by measuring the increase in turbidity at 450 nm, caused by a shift from 4 °C to 37 °C, which allows the in vitro microtubular protein to depolymerize and polymerize, respectively. The assay was carried out in quartz cuvettes containing 1.5 mg/mL microtubular protein and the ligand (except for control cuvette with only DMSO at the same concentration) in a mixture of 0.1 M MES buffer, 1 mM EGTA, 1 mM MgCl_2_, 1 mM β-ME, and 1.5 mM GTP at pH 6.7 (final volume 500 µL). Cuvettes were preincubated at 20 °C for 30 min, to allow ligand binding to tubulin, and subsequently cooled on ice for 10 min. Then, the experiment starts at 4 °C to establish the initial baseline. The assembly process was initiated by a temperature shift to 37 °C and the turbidity produced by tubulin polymerization can be measured by an absorbance increase. After reaching a stable plateau, the temperature was switched back to 4 °C to return to the initial absorption values (to confirm the reversible nature of the monitored process and to determine whether or not the microtubular tubular protein has stabilized). The difference in amplitude between the stable plateau and the initial baseline of the curves was taken as the degree of tubulin polymerization for each experiment. Comparison with control curves under identical conditions but without ligands yielded TPI as a percentage value. All compounds were tested at 10 µM concentration. Compounds that inhibited tubulin polymerization by at least 30% at 10 µM were tested at 20 µM. IC_50_ values were calculated for compounds that inhibit tubulin polymerization by more than 50% at 20 µM. Compounds were dissolved in DMSO, and the final solvent concentration never exceeded 4% (*v*/*v*), which has been reported not to interfere with the assembly process. All the measurements were carried out in at least two independent experiments using microtubular protein from different preparations.

### 4.3. Computational Studies

Docking studies were carried out as previously described [45]. Briefly, dockings were performed in parallel with PLANTS [72] with default settings and 10 runs per ligand and with AutoDock 4.2 [73] runs applying the Lamarckian genetic algorithm (LGA) 100−300 times for a maximum of 2.5 × 10^6^ energy evaluations, 150 individuals, and 27,000 generations maximum. The retrieved poses were automatically assigned to the colchicine subzones using in-house KNIME pipelines [74]. The programs’ docking scores were converted into Z-scores and the poses with best consensus scores were selected as the docking results. Docked poses were analyzed with Chimera [75], Marvin [76], OpenEye [77], and JADOPPT [78].

## Figures and Tables

**Figure 1 ijms-22-01907-f001:**
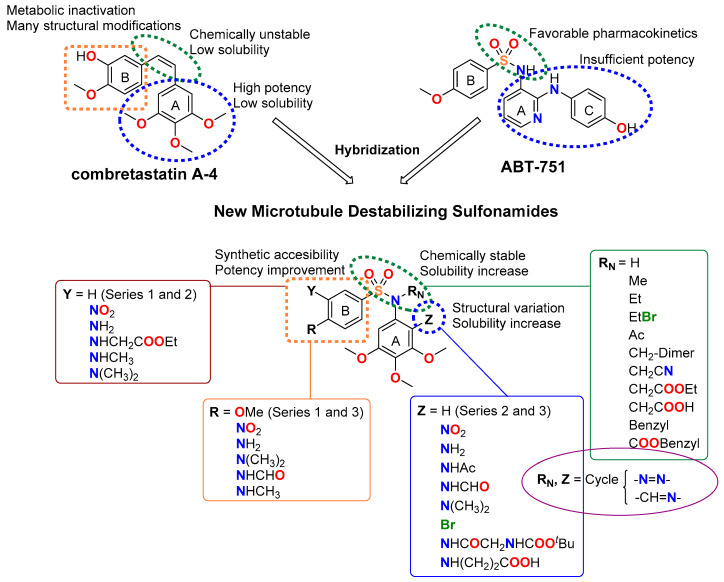
Representative ligands binding at the colchicine site used as a starting point for the rational design of new Microtubule Destabilizing Sulfonamides (MDS). General structure and structure variations of new MDS.

**Figure 2 ijms-22-01907-f002:**
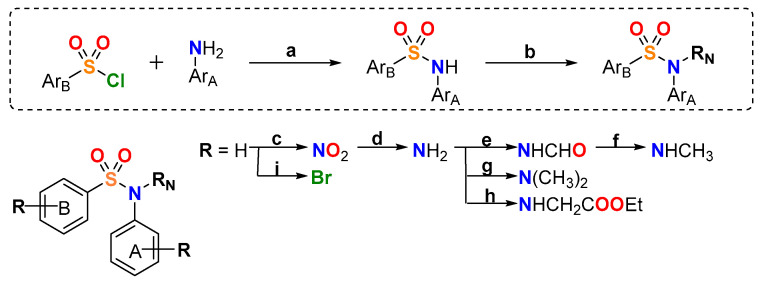
General synthetic approach. Reagents, conditions, and yields: (**a**) Pyridine, CH_2_Cl_2_, rt, 4–8 h, 90–96% (**b**) R_N_ = CH_3_, CH_3_I, KOH, CH_3_CN, rt, 24 h, 63–98%; R_N_ = Ac, acetic anhydride, pyridine, CH_2_Cl_2_, reflux, 8–12 h, 61–83%; R_N_ ≠ CH_3_ and Ac, R_N_-halogen, K_2_CO_3_, dry DMF, rt, 24–48 h, 40–99% (**c**) *tert*-Butyl nitrite, CH_3_CN, 45 °C, 24 h, 60% (**d**) H_2_, Pd/C, EtOAc, rt, 48–72 h, 82–98% (**e**) Formic acid, CH_2_Cl_2_, rt, 24–48 h, 62–82% (**f**) Trichloroacetic acid, NaBH_4_, dry THF, 0°C, 24 h, 97% (**g**) Paraformaldehyde, NaBH_3_CN, AcOH, MeOH, rt, 72–96 h, 95% (**h**) Ethyl 2-bromoacetate, NaI, acetone/THF 1:1, reflux, 48 h, 19% (**i**) NBS, CH_2_Cl_2_, rt, 6 h, 43%.

**Figure 3 ijms-22-01907-f003:**
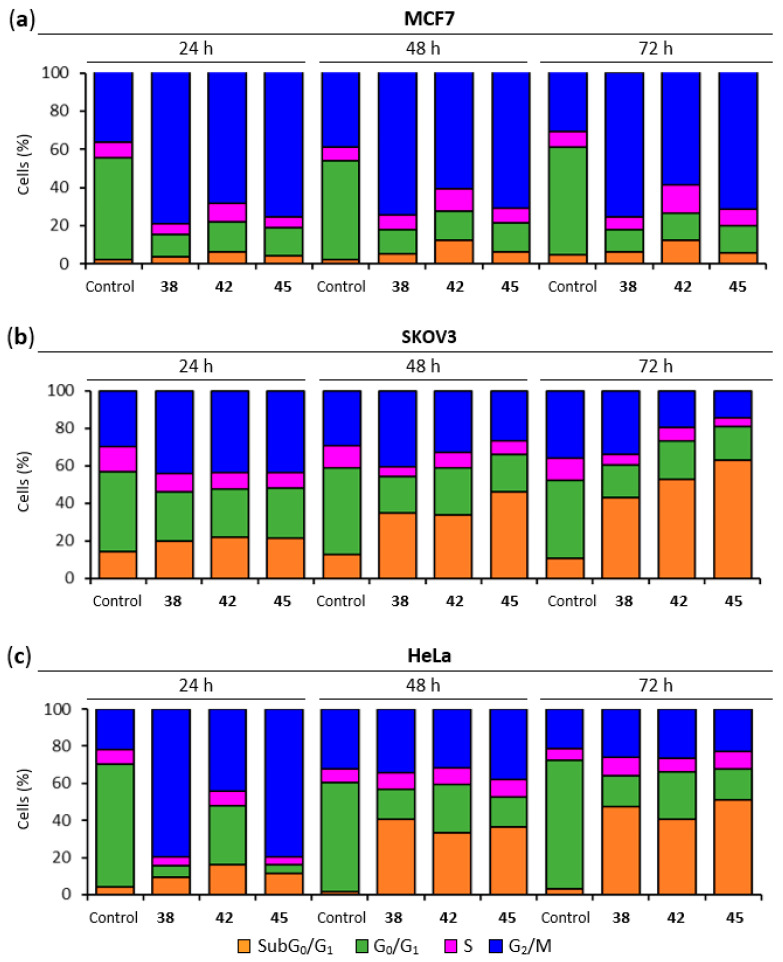
Cell cycle analysis. (**a**) MCF7 cell cycle distribution. Cells were arrested in G_2_/M phase. (**b**) SKOV3 cell cycle distribution. Cells arrested in G_2_/M phase experienced time-dependent cell death (SubG_0_/G_1_ region). (**c**) HeLa cell cycle distribution. Cells were arrested in the G_2_/M phase followed by cell death (SubG_0_/G_1_ region). Cells were incubated with the lead compounds at 600 nM (**38**), 50 nM (**42**), or 400 nM (**45**) for 24, 48, and 72 h, stained with propidium iodide (PI) and their DNA content was analyzed by flow cytometry. Untreated samples were analyzed in parallel. The different cell cycle populations were quantified, expressed in percentages, and represented in bar charts. Data shown are representative of three independent experiments.

**Figure 4 ijms-22-01907-f004:**
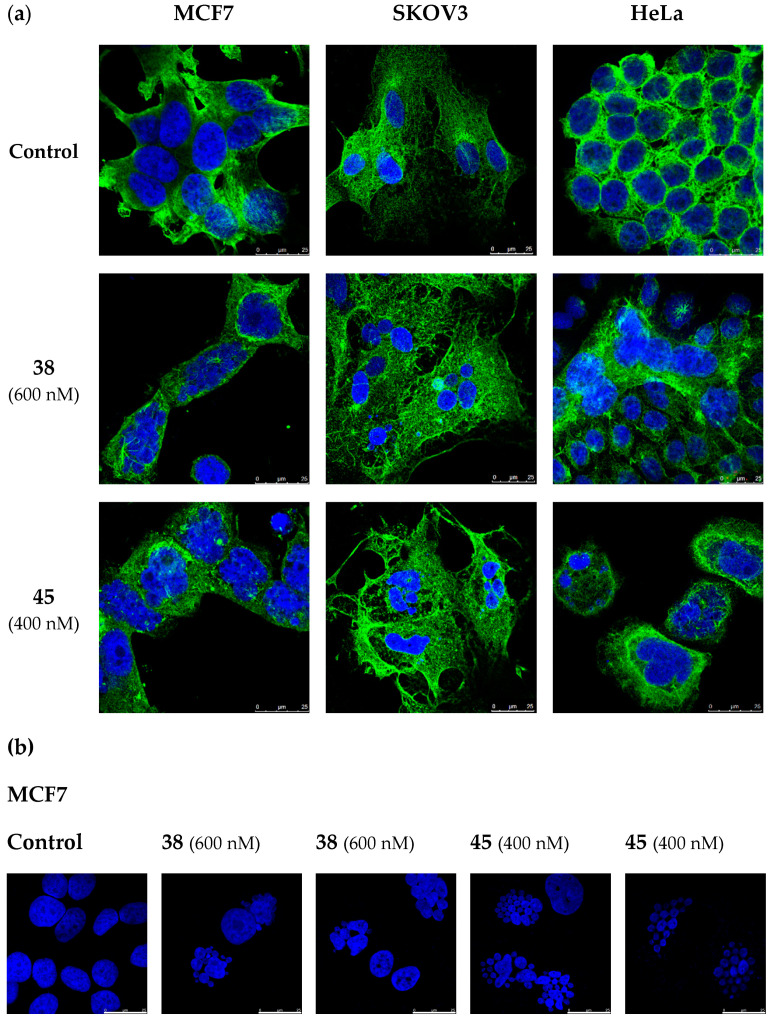
Confocal microscopy experiments in MCF7, SKOV3, and HeLa cells. Cells were incubated in the absence (Control) or the presence of **38** (600 nM) and **45** (400 nM) for 72 h. (**a**) Visualization of the microtubule network. α-Tubulin was stained in green, and nuclei were stained with DAPI (blue fluorescence). (**b**) Multilobulated nuclei (blue) resembling bunches of grapes in MCF7 cells. Photomicrographs are representative of three independent experiments. Scale bar: 25 µm.

**Figure 5 ijms-22-01907-f005:**
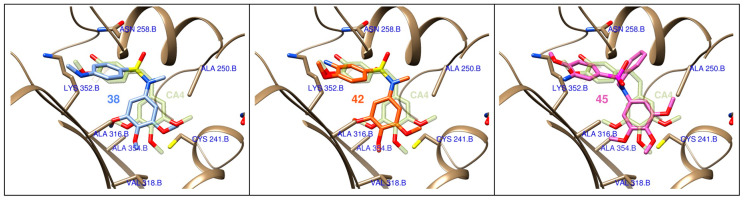
Docking poses for compounds **38**, **42**, and **45** at the colchicine site of tubulin. Combretastatin A-4 (**CA-4**) is shown in diffuse green as a reference.

**Table 1 ijms-22-01907-t001:** Chemical structure, antiproliferative activity against human tumor cell lines, and Tubulin Polymerization Inhibition (TPI) of novel Microtubule Destabilizing Sulfonamides (MDS). Compounds have been divided into three different series according to the substituents on the aromatic B ring (Ar_B_). TM: 3,4,5-trimethoxyphenyl.

**Series 1:** 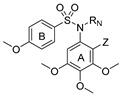		**Antiproliferative Activity** **IC_50_ (nM)**	**TPI**
			**TPI %**	
**Z**	**R_N_**	**Compound**	**HeLa**	**MCF7**	**10 µM**	**IC_50_ (µM)**
H	H	**1a**	240	375	0	>20
H	SO_2–_4-OMePh	**1b**	>1000	>1000	0	>20
H	Me	**2a**	71	127	35	>20
H	CH_2_-Dim	**2b**	>1000	>1000	0	>20
H	Et	**3**	99	87	25	>20
H	EtBr	**4**	94	340	30	>20
H	Ac	**5**	287	270	0	>20
H	CH_2_CN	**6**	143	275	0	>20
H	CH_2_COOEt	**7**	217	335	21	>20
H	CH_2_COOH	**8**	>1000	>1000	0	>20
H	Benzyl	**9**	750	830	21	>20
H	COOBenzyl	**10**	>1000	>1000	11	>20
NO_2_	H	**11**	>1000	>1000	10	>20
NH_2_	H	**12**	>1000	>1000	0	>20
NH_2_	Me	**13**	>1000	>1000	0	>20
NH_2_	CH_2_-Dim	**14**	>1000	>1000	0	>20
NHAc	H	**15**	>1000	>1000	0	>20
NHAc	Me	**16**	>1000	>1000	0	>20
NHCHO	H	**17**	>1000	>1000	4	>20
N(CH_3_)_2_	H	**18**	>1000	>1000	0	>20
N(CH_3_)_2_	Me	**19**	>1000	>1000	0	>20
Br	H	**20**	>1000	>1000	0	>20
Gly-*t*BOC	H	**21**	>1000	>1000	0	>20
Succinic	H	**22**	>1000	>1000	0	>20
-N=N-	**23**	>1000	>1000	0	>20
-N=CH-	**24**	>1000	>1000	0	>20
**Series 2:** 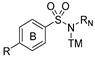		**Antiproliferative Activity** **IC_50_ (nM)**	**TPI**
			**TPI %**	
**R**	**R_N_**	**Compound**	**HeLa**	**MCF7**	**10 µM**	**IC_50_ (µM)**
NO_2_	H	**25**	>1000	>1000	0	>20
NH_2_	H	**26**	>1000	>1000	0	>20
NH_2_	Me	**27**	>1000	>1000	0	>20
N(CH_3_)_2_	H	**28**	230	173	0	>20
N(CH_3_)_2_	Me	**29a**	63	30	40	>20
N(CH_3_)_2_	CH_2_-Dim	**29b**	>1000	>1000	0	>20
N(CH_3_)_2_	Et	**30**	66	81	45	>20
N(CH_3_)_2_	Ac	**31**	183	135	38	14
N(CH_3_)_2_	CH_2_CN	**32**	55	120	17	>20
N(CH_3_)_2_	CH_2_COOEt	**33**	99	91	14	>20
N(CH_3_)_2_	CH_2_COOH	**34**	>1000	>1000	6	>20
N(CH_3_)_2_	Benzyl	**35a**	330	100	72	5.3
N(CH_3_)_2_	COOBenzyl	**35b**	>1000	>1000	0	>20
NHCHO	H	**36**	>1000	>1000	0	>20
NHCH_3_	H	**37**	607	>1000	7	>20
NHCH_3_	Me	**38**	44	61	47	10
**Series 3:** 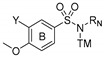		**Antiproliferative Activity** **IC_50_ (nM)**	**TPI**
			**TPI %**	
**Y**	**R_N_**	**Compound**	**HeLa**	**MCF7**	**10 µM**	**IC_50_ (µM)**
NO_2_	H	**40**	>1000	>1000	0	>20
NH_2_	H	**41**	260	96	15	>20
NH_2_	Me	**42**	23	26	41	12
NH_2_	Et	**43**	38	14	37	>20
NH_2_	CH_2_CN	**44**	60	8	5	>20
NH_2_	Benzyl	**45**	25	48	75	3.7
NHCH_2_COOEt	Me	**46**	210	48	13	>20
NHCH_2_COOEt	CH_2_COOEt	**47**	>1000	>1000	0	>20
NHCH_3_	Me	**48a**	440	1190	2	>20
N(CH_3_)_2_	Me	**48b**	>1000	>1000	0	>20
**Paclitaxel** [52]	2.6	2.5	n.d. ^1^	n.d.
**Combretastatin A-4**	2	1	100	3
**ABT-751**	388	180	69	4.4

^1^ n.d.: not determined.

**Table 2 ijms-22-01907-t002:** Aqueous solubility in µg/mL of representative compounds.

Compound	Solub (µg/mL)	Compound	Solub (µg/mL)	Compound	Solub (µg/mL)
**1a**	83	**22**	1690	**37**	43
**1b**	108	**23**	41	**38**	16
**2a**	27	**26**	28	**41**	89
**3**	30	**27**	25	**42**	108
**5**	38	**28**	15	**43**	58
**6**	33	**29a**	7	**44**	46
**11**	88	**29b**	4	**45**	14
**12**	158	**30**	6	**CA-4**	1
**15**	230	**31**	18	**ABT-751**	40
**20**	87	**32**	8	**Paclitaxel** [23,24,25]	<2
**21**	357	**36**	27	–	–

**Table 3 ijms-22-01907-t003:** Antiproliferative activity of the lead compounds and paclitaxel against several human ovarian tumor cell lines.

	Antiproliferative Activity IC_50_ (nM)				
Compound	SKOV3	IGROV-1	A2780	OVCAR-8	OVCAR-3
**38**	46	248	68	74	31
**42**	7	400	42	37	72
**45**	48	492	104	48	58
**Paclitaxel** ^1^	81	39	3	6	17

^1^ IC_50_ values against A2780, OVCAR-8, and OVCAR-3 cell lines from Bicaku, E. et al. evaluation [14].

**Table 4 ijms-22-01907-t004:** Antiproliferative activity against the CA-4 resistant cancer cell line HT-29 and sensitivity of the lead MDS to MDR pumps.

	Antiproliferative Activity IC_50_ (nM)	
Compound	HT-29	HT-29 Verapamil 10 µM
**38**	59	50
**42**	81	79
**45**	300	276
**CA-4**	305	327

**Table 5 ijms-22-01907-t005:** Cell death quantification in Annexin V-FITC/PI double staining experiments. MCF7, SKOV3, and HeLa cells were incubated with the lead compounds for 72 h and/or 6 days, stained with Annexin V-FITC (AnV) and PI, and analyzed by flow cytometry. The results are expressed in percentage as the average of three independent experiments. Untreated samples were analyzed in parallel. Cells were classified into double-negatives (live cells), AnV-positive cells (early apoptosis), double-positives (late apoptosis), and PI-positive cells (necrosis).

	Compound	Annexin V-FITC/PI 72 h (%)				Annexin V-FITC/PI 6 Days (%)			
		Live	EA ^1^	LA ^2^	Necrosis	Live	EA	LA	Necrosis
**MCF7**	**38** (600 nM)	60.5	29.0	5.4	5.1	52.0	45.0	2.7	0.3
	**42** (50 nM)	87.7	8.5	2.3	1.4	79.1	18.9	1.7	0.3
	**45** (400 nM)	70.5	20.1	7.5	1.9	56.7	39.3	3.8	0.3
	**CA-4** (50 nM)	58.3	31.7	3.3	6.7	44.5	52.4	2.7	0.4
	**Control**	93.6	1.8	2.9	1.7	97.4	0.2	0.8	1.6
**SKOV3**	**38** (600 nM)	62.3	30.5	5.4	1.8				
	**42** (50 nM)	85.1	12.2	2.0	0.7				
	**45** (400 nM)	72.7	23.4	2.6	1.3				
	**Paclitaxel** (10 nM)	90.7	3.8	3.1	2.4				
	**Control**	99.2	0.5	0.3	0.1				
**HeLa**	**38** (600 nM)	0.6	1.1	87.6	10.6				
	**42** (50 nM)	7.3	1.4	82.0	9.4				
	**45** (400 nM)	0.8	1.5	84.9	12.7				
	**Control**	90.5	1.7	5.5	2.4				

^1^ Early Apoptosis (EA). ^2^ Late Apoptosis (LA).

## Data Availability

The data that support the findings of this study are available from the corresponding author upon reasonable request.

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
