# Peer review of "Microtubule Destabilizing Sulfonamides as an Alternative to Taxane-Based Chemotherapy"

_ijms, 2021, doi:10.3390/ijms22041907_

Round 1

Reviewer 1 Report

This manuscript describes the synthesis and the biological evaluation of a library of 52 sulfonamide derivatives as anticancer agents The article is well structured. The design is clear, and results look solid and significant. The chemical characterization of the compounds is good. The authors followed and presented a step by step biological screening of the compounds that led to the identification of three lead compounds, namely 38, 42 and 45 both for the antiproliferative and the solubility properties. Docking studies allowed the identification of the key interactions of the leads at the colchicine biding site of tubulin. This manuscript is well written and sounds with a good standard of English language. It includes up-to-date references.

This manuscript is interesting and will be useful to researchers working in the anticancer field.

My only inputs are:

In the abstract: in correspondence to compounds 38, 42 and 45, they should provide the extensive chemical name of each compound, or as well the main core structure and functionalities of the three lead compounds.

Page 6, row 157: revise beter in better

Author Response

Response to Reviewer 1 Comments:

This manuscript describes the synthesis and the biological evaluation of a library of 52 sulfonamide derivatives as anticancer agents The article is well structured. The design is clear, and results look solid and significant. The chemical characterization of the compounds is good. The authors followed and presented a step by step biological screening of the compounds that led to the identification of three lead compounds, namely 38, 42 and 45 both for the antiproliferative and the solubility properties. Docking studies allowed the identification of the key interactions of the leads at the colchicine biding site of tubulin. This manuscript is well written and sounds with a good standard of English language. It includes up-to-date references.

This manuscript is interesting and will be useful to researchers working in the anticancer field.

We thank the reviewer for the supportive comments.

My only inputs are:

Point 1: In the abstract: in correspondence to compounds 38, 42 and 45, they should provide the extensive chemical name of each compound, or as well the main core structure and functionalities of the three lead compounds.

Response 1: We have included the chemical name for compounds 38, 42 and 45 in the abstract. Page 1, rows 35-37.

Point 2: Page 6, row 157: revise beter in better

Response 2: Done. Page 6, row 160.

Reviewer 2 Report

Despite wide application of taxane based chemotherapy to treat a number of human cancers the problem of resistance development and toxic side effects remain a major problem, warranting discovery of new generation of microtubule targeting agents. The problem of resistance development by drug efflux could be addressed by Colchicine site binders like CA-4 and ABT-751 that are in clinical trials. In this manuscript, the authors designed and tested new hybrid Microtubule Destabilising Sulfonamides (MDS) by combining chemical features CA-4 and ABT-751 with the aim to achieve better solubility and potency and at the same time resistance to Pgp mediated efflux. The study is overall well-conceived and executed. I support its publication provided following minor issues are fixed:

  • The authors should show their Tubulin Polymerization Assay data, at least for their lead compounds and a few controls, in a graphical manner in addition to the table.
  • Figure 4: The authors should include CA-4/ABT-751 and/or Taxol controls in their figure. The quality of figures and their representation needs improvement. Figure 4b: two panels are unlabelled. Scale bars are not visible.
  • Page 9, Line 209-212: Reference to supplementary data is missing.
  • Structure analysis: Figure 5 is not legible. I would recommend to label secondary structure elements and amino acid residues with bigger and readable fonts.
  • The authors mention about the Zone 1 of Colchicine site (Line 319) but do not explain in text or show these zones in a figure. This might be confusing for non-specialist reader. I would recommend to either cite the reference paper describing these zone or discuss them in the text.

Author Response

Response to Reviewer 2 Comments:

Despite wide application of taxane based chemotherapy to treat a number of human cancers the problem of resistance development and toxic side effects remain a major problem, warranting discovery of new generation of microtubule targeting agents. The problem of resistance development by drug efflux could be addressed by Colchicine site binders like CA-4 and ABT-751 that are in clinical trials. In this manuscript, the authors designed and tested new hybrid Microtubule Destabilising Sulfonamides (MDS) by combining chemical features CA-4 and ABT-751 with the aim to achieve better solubility and potency and at the same time resistance to Pgp mediated efflux. The study is overall well-conceived and executed.

We thank the reviewer for the supportive comments.

I support its publication provided following minor issues are fixed:

Point 1: The authors should show their Tubulin Polymerization Assay data, at least for their lead compounds and a few controls, in a graphical manner in addition to the table.

Response 1: Following the reviewer suggestion, a figure of tubulin polymerization assays for the most potent compounds, altogether with reference CA-4 and ABT-751 has been included in the supplementary material and referenced in page 12, line 303 and page 18, line 607.

Point 2: Figure 4: The authors should include CA-4/ABT-751 and/or Taxol controls in their figure. The quality of figures and their representation needs improvement. Figure 4b: two panels are unlabelled. Scale bars are not visible.

Response 2: We have improved the quality and representation of Figure 4. The effects of CA-4/ABT-751 and/or Taxol have been extensively published, and we have included references where these results have been already presented. Following the reviewer comment, we have labelled the two panels of Figure 4b that were unlabeled, and we have modified the figures so that the scale bars are clearly visible. Page 11, line 294.

Point 3: Page 9, Line 209-212: Reference to supplementary data is missing.

Response 3: Done. Page 7, line 210.

Point 4: Structure analysis: Figure 5 is not legible. I would recommend to label secondary structure elements and amino acid residues with bigger and readable fonts.

Response 4: Following the reviewer advice, we have labelled the residues in Figure 5 with bigger and more readable fonts, and we think it is now friendlier to the reader. Page 12, line 326.

Point 5: The authors mention about the Zone 1 of Colchicine site (Line 319) but do not explain in text or show these zones in a figure. This might be confusing for non-specialist reader. I would recommend to either cite the reference paper describing these zone or discuss them in the text.

Response 5: We have removed the reference to zone 1 and just kept an indication to the zone behind helix 8, which is sufficient in our opinion to allocate the site using figure 5, even for non-specialists. We hope it is now easier to understand. Page 12, line 322.